# Secreted Phospholipases A2: Drivers of Inflammation and Cancer

**DOI:** 10.3390/ijms252212408

**Published:** 2024-11-19

**Authors:** Ivan Hidalgo, Maria Alba Sorolla, Anabel Sorolla, Antonieta Salud, Eva Parisi

**Affiliations:** 1Research Group of Cancer Biomarkers, Biomedical Research Institute of Lleida (IRBLleida), 25198 Lleida, Spain; 2Department of Medical Oncology, Arnau de Vilanova University Hospital (HUAV), 25198 Lleida, Spain; 3Department of Medicine, University of Lleida, 25198 Lleida, Spain; 4Department of Experimental Medicine, University of Lleida, 25198 Lleida, Spain

**Keywords:** phospholipases, sPLA2, arachidonic acid, tumor microenvironment, EMT

## Abstract

Secreted phospholipase 2 (sPLA2) is the largest family of phospholipase A2 (PLA2) enzymes with 11 mammalian isoforms. Each sPLA2 exhibits different localizations and specific properties, being involved in a very wide spectrum of biological processes. The enzymatic activity of sPLA2 has been well described; however, recent findings have shown that they could regulate different signaling pathways by acting directly as ligands. Arachidonic acid (AA) and its derivatives are produced by sPLA2 in collaboration with other molecules in the extracellular space, making important impacts on the cellular environment, being especially relevant in the contexts of immunity and cancer. For these reasons, this review focuses on sPLA2 functions in processes such as the promotion of EMT, angiogenesis, and immunomodulation in the context of tumor initiation and progression. Finally, we will also describe how this knowledge has been applied in the search for new sPLA2 inhibitory compounds that can be used for cancer treatment.

## 1. Introduction

Phospholipases are enzymes that cleave ester bonds within phospholipids. Lipid products generated in these hydrolytic reactions regulate many different cellular signaling pathways. The distinct enzymes are classified according to the cleaved site of the bond of their phospholipid substrates: the acylhydrolases with phospholipases A1 (PLA1), phospholipases A2 (PLA2), and phospholipases B (PLB); and the phosphodiesterases represented by phospholipase C (PLC) and phospholipase D (PLD) [1]. Among them, PLA2 should be highlighted, which are key enzymes that function as primary generators of fatty acids (FA) and lysophospholipids, precursors of various families of compounds playing multiple roles in inflammation [2]. They are a large superfamily separated into different classes: secreted PLA2 (sPLA2), cytosolic PLA2 (cPLA2), Ca^2+^-independent PLA2 (iPLA2), platelet-activating factor acetylhydrolase PLA2 (PAF-AH PLA2), lysosomal PLA2 (LPLA2), and adipose-tissue-specific PLA2 (AdPLA2). Among them, the first three subfamilies, namely sPLA2, cPLA2, and iPLA2, play critical roles in inflammation and cancer-related diseases, in some cases having cross-reactivity. Therefore, in many cellular contexts, it is impossible to discern the effects of a particular PLA2, since the other family members can influence its activity [3,4].

sPLA2 enzymes comprise the largest family of PLA2, containing 11 mammalian isoforms with a conserved catalytic site. Their cellular effects have been commonly associated with the release of a specific FA, the arachidonic acid (AA), and its eicosanoid metabolites; however, sPLA2 activity also leads to the release of other mono- and polyunsaturated fatty acids (PUFAs) and lysophospholipids, such as lysophosphatidylcholine (LPC) [5]. The variety of phospholipid substrates, the primary and secondary lipid products, and the different known cellular effects of sPLA2s indicate their involvement in a heterogeneity of physiological processes and diseases, including lipid digestion and remodeling, cardiovascular diseases, reproduction, host defense against infections, acute and chronic inflammatory events, and cancer [2,6].

It has been reported that different PLA2 subtypes have an important role in inflammation and are expressed in different cancers, through a mechanism of action yet to be elucidated clearly [7]. Therefore, PLA2 inhibition is considered an advantageous strategy to prevent and treat inflammation and cancer-associated diseases [8]. It seems that PLA2’s functional roles in tumorigenesis are dependent on the enzyme studied, the tissue, and the cancer type involved. Deeper knowledge of the different mechanisms of phospholipase action, as well as the development of new compounds for their inhibition, can give us tools to treat largely prevalent pathologies such as inflammation and cancer.

## 2. Secreted Phospholipases A2

sPLA2 were the first phospholipase A2 enzymes identified and studied in detail in snake venom whose main conserved function is phospholipid hydrolysis [8]. It encompasses a group of 17 enzymes classified according to their chemical structure, containing between 5 and 8 disulphide bonds, a highly conserved Histidine/Aspartic acid catalytic dyad, and a Ca^2+^-binding loop [9]. Several isoforms of phospholipases have been described in viruses, bacteria, fungi, plants, insects, reptiles, and mammals (Table 1) [10], where they display a variety of functions which vary depending on the species source.

In snakes, lizards, and bees, sPLA2 are mostly a compound of their venom whose main conserved function is phospholipid hydrolysis. In particular, sPLA2 penetrate the interphase of the amphipathic structure of the phospholipid head group to catalyze the hydrolysis at the sn-2 position, generating FA and lysophospholipids [36,37]. Moreover, sPLA2 from different venoms (reptiles, insects, and arachnids) demonstrate considerable cytotoxic effects on cancer cells via the induction of apoptosis, cell cycle arrest, and the suppression of proliferation [38]. In plants, sPLA2 is important for the biosynthesis of the plant hormone jasmonic acid, which plays an important role in plant development and defense against pathogens [39,40].

Focusing on human phospholipases, there are 11 isoforms: 10 catalytically active (1B, 2A, 2C, 2D, 2E, 2F, 3, 5, 10, and 12A) and 1 inactive (12B) with a Leucine/Asp in its catalytic site [8]. Phospholipases of groups 1, 2, 5, and 10, also known as classical sPLA2, are closely related enzymes that share its catalytic site, the calcium-binding loop, and up to eight disulfide bonds that give more structural stability. Atypical sPLA2, 3, and 12 only share homology with the other phospholipases in their catalytic site and on the calcium binding loop [41].

The sPLA2 family have a distinct substrate selectivity for sn-2 FA or sn-3 polar head groups. While PLA2G1B, PLA2G2A, and PLA2G2E have the same affinity for the different sn-2 FA, PLA2G5 prefer FA with a lower degree of unsaturation, such as oleic and linoleic acid, and sPLA2 from groups 2D, 2F, 3, and 10 prefer PUFAs such as AA and ω3 docosahexaenoic acid. Furthermore, in relation to the polar head groups, sPLA2G3, 5, and 10 efficiently hydrolyze phosphatidylcholine (PC), while sPLA2 from group 2 hydrolyze phosphatidylethanolamine (PE) much better than PC [5]. In terms of functionality, sPLA2’s specific functions consist of producing lipid mediators, altering membrane phospholipid composition, degrading phospholipids from microorganisms and from the diet, and modifying extracellular non-cellular lipid components such as lipoproteins or microvesicles in response to given microenvironmental signals [5]. Apart from the role of sPLA2 as enzymes, the existence of sPLA2 without enzymatic activity was an early indication that sPLA2 can participate in other physiological settings as ligands for membrane and soluble receptors [9].

One product from PLA2 that has attracted a lot of attention in inflammatory processes and cancer biology is AA, an important polyunsaturated fatty acid which maintains the structure and function of the cell membrane. The first step in the AA cascade is the cleavage and release of AA from the phospholipid bound form [19] (Figure 1). Then, AA can be metabolized by three distinct enzyme systems: cyclooxygenases (COX), lipoxygenases (LOX), and cytochrome P450 (CYP) [42]. The COX pathway is responsible for the conversion of AA into different classes of prostanoids: prostaglandins (PG), prostacyclin D2 (PGD2), prostacyclin I2 (PGI2), and thromboxane A2 (TXA2) [19]. The second metabolic pathway corresponds to the LOX pathway, which catalyzes the deoxygenation of AA into hydroperoxyeicosatetraenoic acids (HpETE). Ultimately, this is followed by the conversion of HpETE to their corresponding hydroeicosatetraenoic acids (HETE), leading to the formation of leukotrienes (LT), lipoxins (LX) and hepoxilins (HO) [20]. Lastly, in the CYP450 pathway, several isoforms of CYP450 catalyze the nicotinamide adenine dinucleotide phosphate-oxidase (NADPH)-depenent conversion of AA to HETE and epoxyeicosatrienoic acids (EET) [21].

An abundant body of work dating back from the 1990s has documented the involvement of sPLA2, specifically PLA2G5, in AA mobilization and attendant eicosanoid production [43]. In general terms, PLA2G5 acts by amplifying the action of cPLA2, which is the key enzyme in the process, via activity-dependent or independent mechanisms. PLA2G5 shows no clear FA preference, and is able to release other fatty acids from cells with regulatory features that are strikingly similar to those of AA release [44]. Moreover, of all members of the sPLA2 family of enzymes, PLA2G5 has been long known to release various fatty acids, including AA and oleic acid, and increases prostaglandin E2 production when added exogenously to phagocytic cells, suggesting the role of this enzyme in inflammation [45]. Furthermore, it is also worth mentioning that although sPLA2 appear to be secreted into the extracellular space after being synthesized inside the cell, compelling evidence has already been provided for their intracellular localization and activities [9], allowing them to participate in AA metabolism, not only in the extracellular space.

Some of PLA2 metabolic byproducts are strongly associated with malignant transformation. One example is prostaglandin E2, a COX metabolite, which possesses the highest tumorigenic and metastatic potential as it inhibits cancer cell apoptosis and increases invasiveness, as well as promoting angiogenesis in tumors [46]. Therefore, there exists strong evidence for the potential of anti-inflammatory agents such as COX inhibitors in cancer prevention.

## 3. sPLA2 and Cancer-Related Inflammation

The concept that inflammation is a critical step in tumor initation is well established. Many cancers arise from sites of infection, chronic irritation, and inflammation. Cells from the tumor microenvironment (TME), like macrophages, neutrophils, and fibroblasts, release growth factors, cytokines, and proteases that modify the cancer niche and become indispensables in the neoplastic process. In addition, tumor cells have integrated some of the signaling molecules of the innate immune system, such as selectins, chemokines, and their receptors for invasion, migration, and metastasis [47,48]. Several factors such as cytokines (Tumor necrosis factor α (TNF-α), interleukins, chemokines, and Transforming growth factor β (TGF-β)), transcription factors (nuclear factor kappa-light-chain-enhancer of activated B cells (NF-κB), STAT3, molecules related to the Wnt and β-catenin pathway), eicosanoids, and kinins that are common effectors in inflammation and cancer are produced in the inflammatory environment [49].

sPLA2 enzymes have been shown to be involved in inflammation by mobilizing pro-inflammatory lipid mediators and anti-inflammatory lipids, as evidenced by various studies [50]. The mobilization of distinct lipids by sPLA2 appears to rely not only on their intrinsic enzymatic properties, but also in the different contexts where sPLA2 act, such as the lipid composition of target membranes or the spatial and temporal availability of downstream lipid-editing enzymes [5]. Products of phospholipases that need a special mention for their relationship with chronic inflammation and cancer are the AA-derived eicosanoids, including PG, LT, and lipoxins (LX). It is well known that eicosanoid levels increase under chronic inflammatory conditions. Granulocytes, macrophages, neutrophils, platelets, mast cells, and endothelial cells are involved in eicosanoid production during inflammation. In this context, eicosanoids will act as pro-inflammatory molecules (such as prostaglandin H2), chemoattractants (LTB4), platelet aggregating factors (TXA2), contractors of smooth muscle (CysLTs), and modifiers of the vascular permeability (LT) [51]. Moreover, the overexpression of COX-2, the enzyme responsible for prostanoid synthesis, both TXA2 and PG, has been strongly related to chronic inflammatory events, tumor growth, angiogenesis, cell invasion, metastasis, and chemoresistance, which lead to a low patient survival rate [52].

The best described sPLA2 related to inflammation and cancer is PLA2G2A. Its expression is induced by pro-inflammatory cytokines and lipopolysaccharides (LPS) [53], and its physiological function consists in the degradation of bacterial membranes [54]. Thus, PLA2G2A is primarily involved in host defense by killing bacteria and triggering innate immunity. Meanwhile, the over-amplification of the response leads to the exacerbation of inflammation by hydrolyzing phospholipids in extracellular microvesicles [55]. Moreover, it has recently been shown that a loss of PLA2G2A leads to an increased expression of pro-inflammatory genes and decreased expression of anti-inflammatory genes in the intestine of mice [56]. The same work highlights the modulation of gut microbiota via changes in the bacterial composition, which influences immune responses and, indirectly, systemic inflammation. Other sPLA2 isoforms expressed in the gut epithelium, such as PLA2G1B and PLA2G10, contribute to the regulation of gut microbiota and systemic responses. These discoveries suggest a broader role for the sPLA2 family in modulating gut microbiota and, indirectly, inflammation [56]. PLA2G3 is a sPLA2 that has been associated with the development of colorectal cancer by the production of pro-inflammatory metabolites, including several lysophospholipids, such as lysophosphatidic acid (LPA) and lysophosphatidylinositol (LPI) [57], which have been suggested to promote colon inflammation through the lysophosphatidic acid 2 (LPA2) and G protein-coupled receptor 55 receptors (GPR55), respectively [58,59].

Another secreted phospholipase that has been studied in inflammation is PLA2G5. It is produced in human and murine macrophages and mast cells, playing a role in AA signaling and being able to act on the outer membrane of cells [60]. Moreover, recent studies showed that PLA2G5 has a strong tendency to hydrolyze phospholipids, with a low content of unsaturated FA at the sn-2 position like dipalmitoyl-phosphatidylcholine. Such FA is an essential component of lung surfactant [5]. Besides the above described roles, PLA2G5 has also been found to function as an anti-inflammatory phospholipase through the modulation of macrophage activity in arthritis [61].

Finally, the secreted phospholipase PLA2G10 has a role in the intestinal epithelium. Apart from that, the overexpression of PLA2G10 in the airway epithelial cells after allergen exposition suggests a role in allergy response. Null mice for PLA2G10 were effective in lowering airway hyper-responsiveness, goblet cell hyperplasia, and pro-asthmatic eicosanoids, which are all important contributing factors to asthma [62]. Moreover, PLA2G10 is overexpressed in M2-like tumor-associated macrophages (TAMs) in the TME in B cell lymphoma. There, it degrades phospholipids in tumor-derived extracellular vesicles, generating immunosuppressive and pro-tumorigenic lipid metabolites. This activity increases IL-10 production and LPA signaling in TAMs that contributes to tumor growth [63]. Overall, PLA2G10 displays dual roles in inflammation. On the one hand, it exerts pro-inflammatory mechanisms in the lungs, and, on the other hand, anti-inflammatory actions in the gut.

## 4. sPLA2 and Cancer

Several studies have revealed that a dysregulated lipid metabolism is one of the fundamental metabolic alterations that enable cancer cell survival and sustain rapid growth and proliferation. It has become clear that changes in FA synthesis, lipolysis, membrane phospholipid hydrolysis, and reacylation pathways are required for cancer cell growth [64]. Different works demonstrate an association between phospholipases and Wnt signaling, β-catenin and the Wnt target gene EphB2, TGF-β and Phosphatidylinositol 3-kinase/Protein kinase B (PI3K/Akt) in different cancer tissues [65,66,67], all of them molecular pathways closely linked to tumor processes. Therefore, sPLA2 have emerged as promising targets in cancer prevention and therapy.

As in inflammatory processes, most of the pathological effects of sPLA2 in cancer are associated with the metabolism of AA and its conversion to eicosanoids [68]. Eicosanoids (PG and LT) can stimulate or promote tumor epithelial cell survival, proliferation, invasion, and metastasis and inhibit apoptosis by modulating multiple signaling pathways. Increased expression of COX-2, one of the enzymes catalyzing the first step of AA conversion into PG, has been associated with a number of malignancies [42,51]. Additionally, the gene knockouts of important enzymes involved in prostanoid metabolism (the cytosolic group 4 of PLA2, COX-1, COX-2, PGE synthase, and prostanoid receptors) lead to reduced tumor growth in mice [51]. Among all the tumor processes where phospholipases play a role, it is worth highlighting the antitumor immune response, angiogenesis, and the epithelial–mesenchymal transition (EMT).

Among sPLA2, group 2A and group 10 enzymes are ubiquitously expressed and are also the most studied sPLA2 in cancer so far [7]. Focusing on the specific case of group 2, the enzymatic activities related to cancer depend on the tissue where the phospholipase is expressed. For instance, they have pro-tumorigenic roles in breast, lung, prostate, ovarian, and esophageal cancers [69,70,71], and conversely, an antitumorigenic activity in gastric and intestinal cancers [66,72].

### 4.1. EMT Driven by Phospholipases

Metastasis is a sequential process, which begins with the EMT process in tumor cells that allows them to acquire a mesenchymal phenotype. During this process, tumor cells lose cell–cell and cell–matrix adhesion and acquire typical mesenchymal features, are able to degrade the extracellular matrix, and show a more mobile and migratory phenotype. This process already occurs physiologically during embryonic development. Therefore, it is logical to think that tumor cells undergoing EMT have stem cell-like properties that make them become highly self-renewing and resistant to the usual anticancer regimens [73]. Actually, recent clinical and preclinical research has provided evidence that cancer progression is being driven not only by a tumor’s underlying genetic alterations, but also by paracrine interactions within the TME [74].

The biological term TME encompasses a wide range of cell types, from cancer cells to non-malignant cells such as the immune cells, fibroblasts, and other components present in the tumor, from blood, lymphatic vascular networks, the extracellular matrix (ECM), and signaling molecules [47]. Several factors in the TME directly induce the occurrence of EMT: inflammatory cytokines, including (TGF-β1), TNF-α, and interleukins, among others [75]. Ultimately, the process is orchestrated by master regulators that coordinate a cascade of events leading to the repression of epithelial genes and the induction of mesenchymal genes. These regulators include the snail zinc-finger family, SNAI1 and SNAI2 (SLUG); the distantly related zinc-finger E-box-binding homeobox family proteins ZEB1 and ZEB2 (SIP1); and the basic helix–loop–helix (bHLH) family of transcription factors, including TWIST1, TWIST2, and E12/E47 [73].

Interestingly, it has been shown that there is a link between EMT and lipid metabolism [76]; however, the specific signaling pathways involved have not yet been fully elucidated (Figure 2). AA metabolites produced by sPLA2 interacting with their own receptors, as EP2 in the cases of PG, or sPLA2 itself acting as a ligand, can function in an autocrine or paracrine manner, activating some signaling pathways [9]. Furthermore, by coupling to its binding proteins (sPLA2-BP), sPLA2 can also be translocated to specific intracellular compartments, such as the cytosol, where they can act as enzymes or receptor ligands and specifically involve themselves in molecular signaling such as decreasing or increasing the permeability of certain ion channels, inhibiting or activating tyrosine kinase receptors, and interfering with integrin-mediated functions, among others [9].

Some studies have explored the crosstalk between the PG produced through the TGF-β pathway and the induction of EMT through COX-2 overexpression [52,77,78]. TGF-β is an anti-inflammatory cytokine and the most potent EMT-inducer. It is produced by cancer cells, myeloid cells, and T lymphocytes and is associated with high-grade malignancies [79].

In addition to the TGF-β pathway, a connection has been seen between the expressions of secretory phospholipases, specifically PLA2G2A, and components of the Wnt signaling pathway [66]. The Wnt pathway is activated by the binding of Wnt ligands to the Frizzled family of membrane receptors, leading to the release and stabilization of β-catenin from the GSK3–AXIN–APC complex. β-catenin then is translocated to the nucleus and becomes part of a transcriptional complex that promotes a gene expression program, which induces the activation of EMT [80].

The promotion of cell growth and proliferation, as well as cell migration and motility via the induction of EMT, is triggered also by several growth factors. To do so, epidermal, fibroblast, insulin, hepatocyte, platelet-derived, and vascular endothelial growth factors act through their cognate tyrosine kinase receptors. Such bindings trigger receptor dimerization, followed by the stimulation of the kinase activity that phosphorylates the receptor and leads to the activation of the PI3K/AKT, ERK/MAPK, p38 MAPK, and JNK pathways [80]. Phospholipases, specifically through AA, can also activate the ERK/MAPK pathway [49,55].

NF-κB is one of the most well-known molecular pathways involved in cancer and has been considered a prototypical pro-inflammatory signaling pathway, activated by cytokines such as interleukin 6 (IL-6) and TNF-α [81,82]. The canonical NF-κB pathway is activated by TNF-α inducing IKKβ phosphorylation to mediate the dissociation of NF-κB from IKBα, resulting in NF-κB translocation to the nucleus, where it activates gene expression. Moreover, in different cancers the existence of an activation of NF-κB signaling via the EGFR/ERK axis promoting snail expression has been shown [82]. The activation of the NF-κB pathway by phospholipase products has been proven by different groups [83]. For instance, Dinicola et al. demonstrated that COX-2/PGE2 axis overexpression induces the activation of the PI3K/Akt pathway, and concomitant NF-kB nuclear translocation, promoting invasiveness in Caco-2 and HCT-8 colon cancer cells [84].

### 4.2. Angiogenesis

Angiogenesis, the formation of new blood vessels from pre-existing vasculature, is a critical tumoral process during cancer growth and metastasis to ensure sufficient blood supply to cover the high nutrient and oxygen demands of tumors for rapid proliferation and survival. Angiogenesis is tightly regulated by a complex interrelation between both pro-angiogenic and anti-angiogenic factors within the TME [85,86].

PLA2G2A and PLA2G2D are two types of sPLA2 enzymes that have gained great attention for their involvement in cancer angiogenesis and metastasis in lung adenocarcinoma and in non-small cell lung cancer (NSCLC), respectively [87,88]. On the one hand, PLA2G2A inhibition has been found to decrease the levels of prostaglandin E2 (PGE2) and proliferation in human lung cancer. PGE2 production is induced by TNF-α in a pro-inflammatory environment. The study of Halpern and colleagues demonstrated that PLA2G2A regulates angiogenesis and metastasis through the production of PGE2, which will, in turn, upregulate the STAT3 transcription factor, activating the expression of ICAM-1, which enhances the invasion of lung cancer cells [88]. Furthermore, PLA2G2D has been shown to enhance the cell viability and angiogenic potential of NSCLC cells [84]. One key mechanism is its ability to modulate the glycolytic pathway in cancer cells. Metabolic reprogramming is a hallmark of cancer cells, where they preferentially utilize glycolysis even in the presence of oxygen, a phenomenon known as the Warburg effect. This metabolic shift provides energy and metabolic intermediates to cancer cells necessary for their rapid growth and proliferation [88,89]. It has been shown that PLA2G2D is silenced due to an upregulation of the glycolytic pathway, a fact that could inhibit aerobic glycolysis in cancer cells. This suggests that PLA2G2D may drive angiogenesis by promoting glycolysis in cancer cells. The upregulation of PLA2G2D has been associated with increased glucose uptake, ATP production, and lactate production, all of which are characteristic features of aerobic glycolysis in cancer cells [90].

Finally, as occurring in other processes, most of the actions of PLA2 on angiogenesis are due to AA and their metabolites, predominantly PG and LT, mainly by enhancing VEGF production [91,92]. Furthermore, enzymes involved in AA metabolism have also been related to angiogenic processes, as in the case of COX [93,94], LOX [95], and CYP450 [96].

### 4.3. Induction of the Immune System

Having seen all the roles of phospholipases in inflammation and tumor processes, it is logical to think that part of their tumoral actions is carried out through the induction of the immune system. Cancer cells can evade the immune system by altering markers and signaling pathways, thereby creating an immunosuppressive TME. This environment is characterized by a dysfunction of antigen-presenting cells (APCs) and the presence of immune regulatory cells, which inhibit T cell priming and suppress cytotoxic T lymphocyte (CTL) function [92].

We have previously described the existence of different members of the sPLA2 family. PLA2G2A is upregulated in lung, prostate, colon, gastric, and breast cancers, favoring tumorigenesis, proliferation, and cell survival; and increasing local inflammation and angiogenesis [7]. Moreover, Miki and collaborators demonstrated that another member of the sPLA2 superfamily, PLA2G2D, acts as an immunosuppressive molecule in skin cancer by increasing the polarization of macrophages towards the M2 phenotype and by diminishing CTL activity [50]. PLA2G10 upregulation is widespread in human cancers and is associated with impaired T cell infiltration into tumor tissues. This overexpression leads to resistance to anti-PD1 immunotherapy by excluding T cells from tumor infiltration [97]. More recently, Ge and collaborators found that PLA2G2A mediates immune escape in pancreatic cancer through its effects on CD8+ T cells: PLA2G2A derived from cancer-associated fibroblasts reduces the secretion of IFN-γ and Granzyme B in CD8+ T cells, impairing their cytotoxic activity against tumor cells and facilitating the immune escape [98].

A role of sPLA has also been found in other inflammatory disorders. PLA2G5 was found in immune cells, including the macrophage cell line P338D1, bone-marrow derived mast cells, T cells, and human neutrophils [99]. It is induced in M2 macrophages and is stimulated by Th2 cytokines IL-4 and IL-13, molecules involved in pathological conditions like asthma [99]. PLA2G2E is expressed at low levels in multiple tissues and is induced in lung epithelial cells and alveolar macrophages following influenza virus infection [100]. Two additional sPLA2 enzymes, PLA2G2D and PLA2G10, are also upregulated in the lungs of mice infected with influenza virus and are implicated in exacerbating infection outcomes [100].

In addition to that, sPLA2 can act indirectly as an immunosuppressive molecule through the synthesis of PGE2. PGE2 is a highly immunosuppressive molecule, which is significantly expressed in colon, lung, breast, and head and neck cancers [101]. Several actions of PGE2 have been described in the immune system, such as inhibiting NK cells, promoting the expansion of regulatory cells, enhancing the proliferation and function of regulatory T cells (Tregs), and also promoting the recruitment of macrophages in TME and stimulating their polarization towards the M2 phenotype by increasing IL-17 expression [92]. Additionally, the endogenous PLA2 inhibitor, Annexin A1, plays a role in cancer progression, contributing to the suppression of the immune response, and indicating cancer aggressiveness [92].

Special attention should be paid to the functions of sPLA2 in macrophages during phagocytosis. Macrophages express surface-associated phospholipases, which may release the AA necessary for endocytosis and electron lucent vesicle formation (ELV). Additionally, PLA2 activity can influence the production of other second messengers indirectly by participating in signaling pathways that involve phospholipase C and phospholipase D, which can generate inositol trisphosphate (IP3) and diacylglycerol (DAG), respectively [102]. Moreover, it has been shown that PLA2G1B and PLA2G10 induce the production of pro-inflammatory cytokines such as TNF-α and IL-6 in human lung macrophages through receptor-mediated mechanisms in which sPLA2 binds to M-type receptors on macrophages, activating the ERK1/2 pathway, leading to cytokine production [103].

## 5. Use of PLA2 Inhibitors to Control Cancer Progression

There are multiple mechanisms where phospholipases induce cancer cell growth and proliferation, either through the metabolism of the AA or by releasing lysophospholipids, as has previously been explained [4]. For this reason, researchers have explored the potential benefits of targeting sPLA2 in cancer treatment. However, it should be noted that whilst some clinical trials have been conducted, very few PLA2 inhibitors have been approved for the treatment of cancer. In addition, some PLA2 inhibitors with anticancer activity exert their antitumoral function not directly by inhibiting PLA2 but by targeting other crucial signaling pathways deregulated in cancer.

### Inhibitors of sPLA2

Several sPLA2 inhibitors have been developed, including specific inhibitors, those that differentiate between the catalytic serine and histidine classes of PLA2, and those that can differentiate between individual PLA2 isoforms within the same class [4] (Table 2).

The class of sPLA2 with the larger number of inhibitors described to date is PLA2G2A. For example, one such inhibitor is LY311727, which is an indole derivative and was developed by Eli Lilly laboratories [104]. It was initially tested to inhibit the invasion of Toxoplasma gondii, and later on, resulted in a significant decrease in the ability of human oral squamous cell carcinoma cells to form spheres [115]. Varespladib (LY315920) is another indole-based nonspecific pan-secretory PLA2 inhibitor that potently inhibits mammalian PLA2G2A, PLA2G5, and PLA2G10. It has been studied for its anticoagulant effect on PLA2 toxins from snake venoms [105]. The favorable properties of varespladib on lipid and inflammatory markers encouraged the study of the effects of this compound in cardiovascular disease outcomes. Unfortunately, varespladib did not reduce the risk of recurrent cardiovascular events and significantly increased the risk of myocardial infarction in a phase III clinical trial [116]. Recent findings suggest the antitumoral activity of varespladib in Epstein–Barr-induced lymphoma. The intrasplenic administration of varespladib in humanized mice with this lymphoma suppressed its formation [63]. Similarly, darapladib, the most selective inhibitor of lipoprotein-associated PLA2 (also named PLA2G7), represents a novel class of therapeutic agents that target inflammation to treat high-risk atherosclerosis [106]. In the context of cancer, darapladib sensitizes cancer cells to ferroptosis by remodeling lipid metabolism [117]. Phase III clinical trials evaluating darapladib showed the efficacy of the inhibitor in blocking atherosclerosis with an absence of severe adverse events. However, there are still no trials assessing the administration of darapladib for cancer therapy.

A family of potent sPLA2 inhibitors are thielocins, secondary metabolites from the fungi Thielavia terricola [118]. Thielocin B1 is a potent inhibitor of PLA2G2 [107] and has also been described to inhibit the proteasome assembling chaperone (PAC) complexes [119], being previously discovered as an inhibitor of a protein–protein interactions for TCF7/β-catenin, PAC1/PAC2, and the PAC3 homodimer from a library containing 123.599 samples [120]. Nevertheless, further research evaluating the antitumor activity of thielocin B1 is still to be performed.

Other functional groups proved to inhibit sPLA2 are sulfonamides [121]. Sulfonamides have functionally been modified, adding trifluoromethyl groups to be more active against sPLA2 [108]. Dabrafenib is a good example of a sulfonamide compound with fluoride groups, known to inhibit kinase activity, specially targeting the MAPK pathway. Dabrafenib has been approved by the Food and Drug Administration (FDA) for the treatment of BRAF mutated (V600E) advanced melanoma, and recently approved in combination with trametinib for BRAF mutated advanced solid tumors [122]. In addition, dafrafenid is claimed to be repositioned for the treatment of LPS-mediated vascular inflammatory responses, harboring anti-inflammatory properties upon treatment on LPS-activated human umbilical vein endothelial cells and mice [123].

Carboxamines, such as N-Benzyl-4,6-difluoro-1H-indole-2-carboxamide, act as selective PLA2G10 inhibitors [124], and some related compounds have been proven to demonstrate anti-cancer activity, especially in pediatric brain cancer cell lines [109]. Specifically, some indole-2-carboxamide derivatives have the potential to reduce cell viability and cell proliferation of KNS42, BT12, BT16, and DAOY human cell lines, derived from glioblastoma, teratoid/rhabdoid tumors, and medulloblastoma, respectively, without affecting the non-neoplastic human fibroblast cell lines (HFF1).

Triterpenoids such as celastrol, maslinic acid, oleanolic acid, and ursolic acid are documented to inhibit sPLA2 [125]. All of them have antitumor properties both in vivo and in vitro, modulating oncogenic signaling pathways rather than directly inhibiting sPLA2 [110,111,112].

Venomous and nonvenomous snakes have PLA2 inhibitory proteins in their blood serum called phospholipase inhibitors or PLI [126]. The most abundant PLI family are γPLI, which appear to be specific for PLA2 1, 2, and 3 groups. One example is γCdcPLI, a glycoprotein isolated from Crotalus durissus collilineatus, which has been proposed to inhibit both sPLA2 and/or cPLA2. Interestingly, γCdcPLI demonstrated antitumoral effects by inhibiting the PI3K/Akt pathway on breast cancer cell lines [114].

Whilst targeting sPLA2 seems promising, there are some challenges to consider. As mentioned previously, and given that sPLA2 is involved in various physiological processes in different tissues and organs, its inhibition may result in unwanted side effects. In addition, the effectiveness of these inhibitors may greatly vary among different cancer types. Thus, developing a selective sPLA2 inhibitor which does not interfere with nontumoral cells could be highly challenging and at the same time an important issue to be considered for the development of new drugs for cancer treatment.

## 6. Conclusions

In this review, we have provided an overview of the biological functions of sPLA2 mainly focused on cancer and cancer-related processes, such as inflammation, EMT, angiogenesis, and antitumoral immune response. PLA2 are a large family of phospholipase enzymes with a great variety of phospholipid substrates, which have several cellular effects and are involved in a heterogeneity of physiological processes and diseases [2,6]. In mammalian cells, there are 11 isoforms with a conserved catalytic site, and their expression varies between tissues [5].

Various studies evidenced that sPLA2 enzymes promote inflammation by mobilizing pro-inflammatory lipid mediators and anti-inflammatory lipids [22], and that some of the metabolites resulting from their action, such as eicosanoids, increase during chronic inflammatory conditions [51]. It is also well known that dysregulated lipid metabolism is one of the fundamental metabolic alterations that enables cancer cell survival and sustained rapid cancer cell growth and proliferation [64]. Phospholipases and their main product, AA, have been shown to play roles in several tumoral processes, such as the promotion of EMT, angiogenesis, and the induction of the immune system [42]. For the reasons exposed, sPLA2 have emerged as promising targets in cancer prevention and therapy. In fact, several sPLA2 inhibitors have been developed for cancer treatment [4], such as darapladib, which sensitizes cancer cells to ferroptosis [117]. However, further research regarding the antitumoral activity and selectivity of these compounds is warranted.

## Figures and Tables

**Figure 1 ijms-25-12408-f001:**
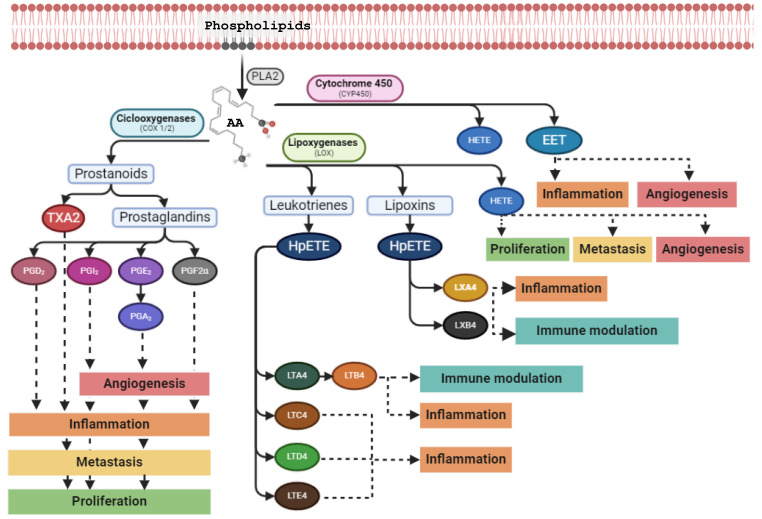
Metabolic products of arachidonic acid (AA). AA is released from membrane phospholipids by phospholipase A2 (PLA2) and metabolized by three main enzyme families: cyclooxygenases (COX), lipoxygenases (LOX), and cytochrome P450 (CYP450). COX-1 and COX-2 convert AA into prostaglandins (PG) D, I, E, A, and F2α, and thromboxanes A2 (TXA2). LOX catalyze the formation of hydroperoxyeicosatetraenoic acids (HpETE) and hydroxyeicosatetraenoic acids (HETE), leukotrienes (LT) A, B, C, D, and E, and lipoxins (LX) A and B. Lastly, CYP450 converts AA in epoxyeicosatrienoic acids (EET) and HETE. Dashed arrows represent each metabolite’s role in inflammation, angiogenesis, metastasis, proliferation, and immune modulation.

**Figure 2 ijms-25-12408-f002:**
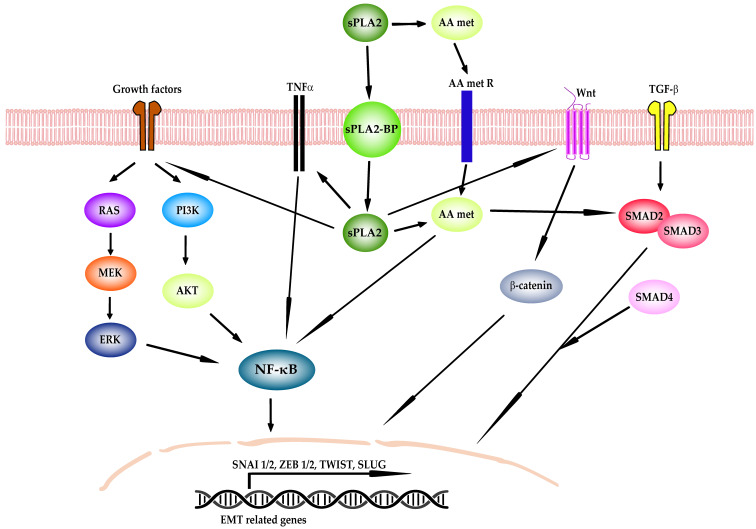
Signaling networks that regulate epithelial–mesenchymal transition (EMT) activated by secreted phospholipases A2 (sPLA2). sPLA2 can activate EMT signaling pathways directly by engaging with sPLA2A-binding proteins (sPLA2A-BP) and being translocated to the cytoplasm, or by producing arachidonic acid and downstream metabolites (AA met). Transforming growth factor β (TGF-β), growth receptors and tumor necrosis factor α (TNFα) signaling pathways can induce EMT by the activation of the transcription factors (TF) SNAI1, ZEB1/2, TWIST, and SLUG. TGF-β induces EMT by the phosphorylation of Smad2 and Smad3, which localize to the nucleus with Smad4 to activate EMT TF. Several growth factors that act through tyrosine kinase receptors, such as epidermal growth factor (EGF), fibroblast growth factor (FGF), and hepatocyte growth factor (HGF), promote EMT thought the RAS-Mitogen-activated protein kinase (MAPK)/ERK signaling cascade or the Phosphatidylinositol 3-kinase/Protein kinase B (PI3K)/Akt axis, which ultimately activate nuclear factor kappa-light-chain-enhancer of activated B cells (NF-κB). TNFα also activates the NF-κB pathway. Finally, Wnt stabilizes β-catenin, which translocates to the nucleus to activate ZEB1 and SNAI1 directly.

**Table 1 ijms-25-12408-t001:** Classification of the different groups of secretory phospholipases A2.

sPLA2	First Source	Tissue	Molecular Weight (kDa)	Aa	Signal Peptide (Aa)	UniProtKB Source	Refs.
**1A**	Serpents	Venom	13–15	119	-	P15445 (cobra)	[11,12]
**1B**	Mammals	Intestinal tract, lungs, pancreas	16	148	1–15	P04054 (human)	[12,13]
**2A**	Serpents, Mammals	Venom, Synovial fluid, liver, tongue, prostate, spleen, intestinal tract	16	138 144	- 1–20	A0A8C6Y0K5 (cobra) P14555 (human)	[14,15,16,17,18]
**2B**	Serpents	Venom	13–15	118	-	P00620 (gaboon viper)	[11]
**2C**	Mammals	Testis, endometrial	15–16	149	1–18	Q5R387 (human)	[11,19]
**2D**	Mammals	Pancreas, spleen, umbilical cord blood	14–15	145	1–20	Q9UNK4 (human)	[12,17,20,21]
**2E**	Mammals	Brain, heart, uterus	14–15	142	1–19	Q9NZK7 (human)	[10]
**2F**	Mammals	Testis, embryo, thymus, spleen, synovial fluid, liver, prostate	16–17	168	1–20	Q9BZM2 (human)	[20,21,22,23]
**3**	Insects, Arachnids, Reptiles, Mammals	Venom, brain, immune cells	15–57	167 509	1–18 1–19	P00630 (bee) Q9NZ20 (human)	[20,24]
**5**	Mammals	Heart, lung, immune cells, embryo	14–16	138	1–20	P39877 (human)	[25,26,27,28]
**6**	Serpents	Venom	15–16	138	1–16	Q6H3C8 (Chinese viper)	[29]
**7**	Serpents	Venom	15	138	1–16	P70089 (Indian viper)	[30]
**9**	Marine Snails	Venom	8	77	-	Q9TWL9 (marine snail)	[20,31]
**10**	Mammals	Heart, spleen, colon, thymus, lungs, nervous system, immune cells	14–18	165	1–31	O15496 (human)	[20,32]
**11**	Plants	Sprout	12–13	138	1–21	Q9XG80 (rice)	[10]
**12A**	Mammals	Uterus, heart, skeletal muscle, kidney, liver, pancreas	21	189	1–22	Q9BZM1 (human)	[33,34]
**12B**	Mammals	Liver, small intestine, kidney	21	195	1–19	Q9BX93 (human)	[33,34,35]

**Table 2 ijms-25-12408-t002:** Inhibitors of phospholipases.

Chemical Group	Compound	Selectivity	Empirical Formula	Molecular Weight (Da)	Cas Number	Ref.
**Indole derivatives**	LY311727	Inhibits sPLA2 IIA	C_22_H_27_N_2_O_5_P	430.43	164083-84-5	[104]
	Varespladib	Inhibits sPLA2-IIA, and less efficiently sPLA2-V and sPLA2-X	C_21_H_20_N_2_O_5_	380.39	172732-68-2	[105]
	Darapladib	Inhibits LpPLA2 (or sPLA2 VII)	C_36_H_38_F_4_N_4_O_2_S	666.77	356057-34-6	[106]
**Thielocins**	Thielocin B1	Inhibits sPLA2-II	C_53_H_58_O_17_	967.02	144118-26-3	[107]
**Sulfonamides**	Dabrafenib	Inhibits sPLA2-IIA	C_23_H_20_F_3_N_5_O_2_S_2_	519.6	1195765-45-7	[108]
**Carboxamines**	1H-indole-2-carboxamide	Inhibits sPLA2-X	C_9_H_8_N_2_O	160.17	1670-84-4	[109]
**Triterpenoids**	Celastrol	Inhibits sPLA2-IIA	C_29_H_38_O_4_	450.61	34157-83-0	[110]
	Maslinic acid	Inhibits sPLA2-IIA	C_30_H_48_O_4_	472.7	4373-41-5	[111]
	Oleanolic acid	Inhibits sPLA2-II	C_30_H_48_O_3_	456.7	508-02-1	[112]
	Ursolic acid	Inhibits sPLA2-IIA	C_30_H_48_O_3_	456.7	77-52-1	[113]
**PLI**	γCdcPLI	Inhibits sPLA2 and/or cPLA2	-	22,340	-	[114]

## Data Availability

No new data were created or analyzed in this study.

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
