# Peer review of "Secreted Phospholipases A2: Drivers of Inflammation and Cancer"

_ijms, 2024, doi:10.3390/ijms252212408_

Round 1
Reviewer 1 Report
Comments and Suggestions for Authors
The review form Parisi’s group deals with the secreted phospholipases A2 as drivers of inflammation and cancer. The abstract is quite weak and seems to be written superficially, with phrases such as “being involved in various biological events, such as having an acting role in…”. The introduction fails to present on overview of the PLA2 family of enzymes. Thus, reference 2 is about secretory PLA2 while reference 3 is 20 years old and focused only on PLA2 as anticancer drugs. The only (more recent) reference is reference 4, which is 4 years old, and then older reviews are cited. Did the authors do the recent literature on PLA2s? This is quite important since many inhibition studies on sPLA2 failed because they focused on this subfamily of PLA2s, neglecting the others, and thus generating significant side effects, which ultimately led to the termination of the research programs. Any review on a subclass of PLA2 should contain a description of the main classes of PLA2, their cellular localization and biological effects, as any sPLA2 inhibitor cannot avoid these related isozymes in vivo. This was acknowledged by the authors too at the end of intro, without going into any details.
The confusion continues in section 2, where it was stated that “sPLA2 was the first phospholipase A2 enzyme identified and studied in detail in snake venom, mammalian pancreas and mammalian cells”, when anybody familiar with the field knows that these sources contain different sPLA2s (IB, IIA, IA) as presented in fact by authors in Table 1. After 3 paragraphs that were relatively ok, the authors start to detail the effects of PLA2 on COX-related inflammation pathway, without specifying that sPLA2 are secreted enzymes that act primarily extracellularly. Yes, they can intervene in the COX pathway but only in very specific cases and their action should not be confused or mixed with cPLA2 action. At this point the reviewer raises again the requirement of a thorough knowledge of the other PLA2s, their localization, mechanism of action and interference with sPLA2. The authors are encouraged to refer to (a) specific sPLA2(s) when they associate it with a disease and avoid generic statements such as “sPLA2 involvement in X”. The whole section 4 and 4.1 needs to be revised as it is very confusing and authors are mixing the effects of different classes of PLA2s. Sections 4.2 and 4.3 are better written, linking specific sPLA2s with their biological action. These sections should serve as a model for the whole . The inhibitor section has to improved also, specifying the selectivity of the enumerated inhibitors on other PLA2 classes (see above).
Comments on the Quality of English Language
see comments to authors
Author Response
We would like to acknowledge the reviewer for the suggestions. I will reply point by point to all the requests.
The review form Parisi’s group deals with the secreted phospholipases A2 as drivers of inflammation and cancer. The abstract is quite weak and seems to be written superficially, with phrases such as “being involved in various biological events, such as having an acting role in…”.
We thank the reviewer for the suggestion. In this regard, we have modified the abstract, resulting in the following:
“Secreted phospholipase 2 (sPLA2) is the largest family of phospholipase A2 (PLA2) enzymes with 11 mammalian isoforms. Each sPLA2 exhibits different localizations and specific properties, being involved in a very wide spectrum of biological processes. The enzymatic activity of sPLA2 has been well described, however, recent findings have shown that they could regulate different signaling pathways by acting directly as ligands. Arachidonic acid (AA) and its derivatives are produced by sPLA2 in collaboration with other molecules in the extracellular space producing important impacts on the cellular environment, being especially relevant in contexts of immunity and cancer. For these reasons, this review focuses on sPLA2 functions in processes such as the promotion of EMT, angiogenesis and immunomodulation in the context of tumor initiation and progression. Finally, we will also describe how this knowledge has been applied in the search for new sPLA2 inhibitory compounds that can be used for cancer treatment.”
The introduction fails to present on overview of the PLA2 family of enzymes. Thus, reference 2 is about secretory PLA2 while reference 3 is 20 years old and focused only on PLA2 as anticancer drugs. The only (more recent) reference is reference 4, which is 4 years old, and then older reviews are cited. Did the authors do the recent literature on PLA2s? This is quite important since many inhibition studies on sPLA2 failed because they focused on this subfamily of PLA2s, neglecting the others, and thus generating significant side effects, which ultimately led to the termination of the research programs. Any review on a subclass of PLA2 should contain a description of the main classes of PLA2, their cellular localization and biological effects, as any sPLA2 inhibitor cannot avoid these related isozymes in vivo. This was acknowledged by the authors too at the end of intro, without going into any details.
We fully agree with the reviewer's comments and are fully aware that there is a huge variety of different classes of phospholipases and their effects are cross-related. However, our review focuses on the effects of phospholipases on inflammation and related carcinogenic processes, so we preferred to concentrate our work on secretory phospholipases, which are the ones commonly associated with these processes. For this reason and because there is plenty of reviews about the whole family, we only mention the different classes of phospholipases very briefly, focusing only on the secretory ones.
However, we have tried to improve the introduction paragraph with more information about the PLA2 family:
“They are a large superfamily separated into different classes: secreted PLA2s (sPLA2s), cytosolic PLA2s (cPLA2s), Ca2+-independent PLA2s (iPLA2s), platelet-activating factor acetylhydrolase PLA2s (PAF-AH PLA2s), lysosomal PLA2s (LPLA2s) and adipose-tissue-specific PLA2s (AdPLA2s). Among them, the first three subfamilies, namely sPLA2, cPLA2 and iPLA2, play critical roles in inflammation and cancer-related diseases in some cases having cross-reactivity. Therefore, in many cellular contexts it is impossible to discern the effects of a particular PLA2, since the other family members can influence its activity.”
It is also worth noting that many of the articles we have used in the introduction are old is due to the fact that most of the characterization work on the phospholipase family was carried out 20 years ago. The most recent articles on the subject refer to these previous studies when describing the characteristics of these enzymes, so we consider that whenever possible we should cite the original articles.
The confusion continues in section 2, where it was stated that “sPLA2 was the first phospholipase A2 enzyme identified and studied in detail in snake venom, mammalian pancreas and mammalian cells”, when anybody familiar with the field knows that these sources contain different sPLA2s (IB, IIA, IA) as presented in fact by authors in Table 1.
Thank you for the suggestion, to avoid misunderstandings we have changed the sentence to “sPLA2 were the first phospholipase A2 enzymes identified and studied in detail in snake venom whose main conserved function is phospholipid hydrolysis ”
After 3 paragraphs that were relatively ok, the authors start to detail the effects of PLA2 on COX-related inflammation pathway, without specifying that sPLA2 are secreted enzymes that act primarily extracellularly. Yes, they can intervene in the COX pathway but only in very specific cases and their action should not be confused or mixed with cPLA2 action.
Thanks for the suggestion. To clarify this, we have added an extra paragraph to the section with:
“An abundant body of work dating back from the 90's has documented the involvement of sPLA2, specifically PLA2G5, in AA mobilization and attendant eicosanoid production (Balestrani 2006). In general terms, PLA2G5 acts by amplifying the action of cPLA2, which is the key enzyme in the process, via activity-dependent or-independent mechanisms. PLA2G5 shows no clear FA preference, and is able to release other fatty acids from cells with regulatory features that are strikingly similar to those of AA release (Astudillo 2019). Moreover, of all members of the sPLA2 family of enzymes, PLA2G5 is long known to release various fatty acids including AA and oleic acid, and increases prostaglandin E2 production when added exogenously to phagocytic cells, suggesting a role for this enzyme in inflammation (Bezzine 2000). Furthermore, it is also worth mentioning that although sPLA2 appear to be secreted into the extracellular space after being synthesized inside the cell, compelling evidence has already been provided for their intracellular localization and activities (Ivanusec 2022), allowing them to participate in AA metabolism not only in the extracellular space.”
At this point the reviewer raises again the requirement of a thorough knowledge of the other PLA2s, their localization, mechanism of action and interference with sPLA2. The authors are encouraged to refer to (a) specific sPLA2(s) when they associate it with a disease and avoid generic statements such as “sPLA2 involvement in X”. The whole section 4 and 4.1 needs to be revised as it is very confusing and authors are mixing the effects of different classes of PLA2s. Sections 4.2 and 4.3 are better written, linking specific sPLA2s with their biological action. These sections should serve as a model for the whole.
Thank you for the suggestion. It is worth mentioning, however, that in many of the articles we have reviewed, the specific phospholipase performing the function is not mentioned, but instead the effects of secretory phospholipases in general are discussed. Furthermore, some functions described in this section are performed by arachidonic acid metabolites produced by various phospholipases depending on the tissue studied. However, wherever possible, we have indicated which enzyme performs the function. Despite all this and in order to improve the section, we added an additional paragraph explaining how the signaling pathways are activated by sPLA:
“AA metabolites produced by sPLA2 interacting with their own receptors, as EP2 in the cases of PG, or sPLA2 itself acting as ligands can function in an autocrine or paracrine manner activating some signaling pathways (Ivanusec 2022). Furthermore, by coupling to its binding proteins (sPLA2-BP), sPLA2 can also be translocated to specific intracellular compartments, such as the cytosol, where they can act as enzymes or receptor ligands and specifically involve themselves in molecular signaling such as decreasing or increasing the permeability of certain ion channels, inhibiting or activating tyrosine kinase receptors, interfering with integrin-mediated functions, among others (Ivanusec 2022)”.
The inhibitor section has to improved also, specifying the selectivity of the enumerated inhibitors on other PLA2 classes (see above).
Following the reviewer’s advice, we have added an additional column in Table 2 specifying the selectivity of the listed inhibitors described in the section.
Reviewer 2 Report
Comments and Suggestions for Authors
The aim of this review is to describe the most recent findings regarding the role of secreted phospholipases in physiological and pathological conditions related to immunity and cancer. It also aims to describe the current knowledge regarding secreted phospholipase inhibitors and their use as anticancer therapies.
The review helps to focus on the role of sPLA2 in cancer and to get an idea of the main molecules that have been studied to control the enzymatic activity of these proteins. I therefore believe that the manuscript should be accepted for publication in this journal, however, there are several points that should be improved.
In the introduction, the authors describe the class of phospholipase enzymes, and then specify those of class A2. The first paragraph is dedicated to the description of secreted phospholipases of type A2, summarized in Table 1. Table 1, which also shows the AA number of the proteins reported as an example, should specify how much of the AA number includes the signal peptide and when it does not.
Riga 81, pag 3: the sentence ‘sPLA2 with the polar head, group 3, 5 and 10 hydrolyze phosphatidylcholine (PC), whereas PLA2G2 has higher affinity for phosphatidylethanolamine (PE)’ must be rewritten.
Figure 1, describing the arachidonic acid transformation pathway, is a classic figure, found in any biochemistry textbook and does not add anything particular to the review. Authors should enrich and/or modify it to provide more specific information relevant to the review, for example by highlighting the role of the various derivatives in the diseases they describe.
Minor: To the abbreviations of phospholipase genes ex PLA2G2A, PLA2G2D etc the 's' should not be added to specify that they are secreted because they are exclusively secreted phospholipases
In the paragraphs 3 and 4, where the role of sPLA2 in inflammation and cancer is described, the authors focus on the enzymatic activity of proteins. Regard the protein-protein interaction activity of sPLA2 they report only the interaction between sPLA2 and M-type receptor. However, there are other relevant interactions to report (with integrins, with EGFR for example) that may play a role in the processes described.
Figure 2 illustrating the ‘Signaling networks that regulate Epithelial Mesenchymal Transition (EMT) activated by phospholipases’ should also highlight how these signaling pathways are activated by sPLA2s.
Round 2
Reviewer 1 Report
Comments and Suggestions for Authors
The revised version of the manuscript is improved and the article can be published in the new form.
Reviewer 2 Report
Comments and Suggestions for Authors
The authors took my suggestions, I think the manuscript has now improved and contains more information and is more focused.